# Identification of PTPN12 Phosphatase as a Novel Negative Regulator of Hippo Pathway Effectors YAP/TAZ in Breast Cancer

**DOI:** 10.3390/ijms25074064

**Published:** 2024-04-05

**Authors:** Sahar Sarmasti Emami, Anni Ge, Derek Zhang, Yawei Hao, Min Ling, Rachel Rubino, Christopher J. B. Nicol, Wenqi Wang, Xiaolong Yang

**Affiliations:** 1Department of Pathology and Molecular Medicine, Queen’s University, Kingston, ON K7L 3N6, Canada; saharsarmasti1990@gmail.com (S.S.E.); 21ag87@queensu.ca (A.G.); derek.zhang@queensu.ca (D.Z.); yh5@queensu.ca (Y.H.); 17ml35@queensu.ca (M.L.); rubinor@queensu.ca (R.R.); nicolc@queensu.ca (C.J.B.N.); 2Department of Developmental and Cell Biology, University of California at Irvine, Irvine, CA 92617, USA; wenqiw6@uci.edu

**Keywords:** Hippo pathway, breast cancer, YAP, TAZ, phosphatase, PTPN12, tumor suppressor, cell proliferation, cell migration

## Abstract

The Hippo pathway plays crucial roles in governing various biological processes during tumorigenesis and metastasis. Within this pathway, upstream signaling stimuli activate a core kinase cascade, involving MST1/2 and LATS1/2, that subsequently phosphorylates and inhibits the transcriptional co-activators YAP and its paralog TAZ. This inhibition modulates the transcriptional regulation of downstream target genes, impacting cell proliferation, migration, and death. Despite the acknowledged significance of protein kinases in the Hippo pathway, the regulatory influence of protein phosphatases remains largely unexplored. In this study, we conducted the first gain-of-functional screen for protein tyrosine phosphatases (PTPs) regulating the Hippo pathway. Utilizing a LATS kinase biosensor (LATS-BS), a YAP/TAZ activity reporter (STBS-Luc), and a comprehensive PTP library, we identified numerous novel PTPs that play regulatory roles in the Hippo pathway. Subsequent experiments validated PTPN12, a master regulator of oncogenic receptor tyrosine kinases (RTKs), as a previously unrecognized negative regulator of the Hippo pathway effectors, oncogenic YAP/TAZ, influencing breast cancer cell proliferation and migration. In summary, our findings offer valuable insights into the roles of PTPs in the Hippo signaling pathway, significantly contributing to our understanding of breast cancer biology and potential therapeutic strategies.

## 1. Introduction

The Hippo pathway, initially identified in *Drosophila* as a novel regulator of animal size [1,2,3,4,5,6,7,8,9], has emerged as a signaling pathway with a central role in diverse biological processes, including tissue homeostasis [1,2,3,4,5,6,7,8,9], tumorigenesis [10,11], immune response [12], mechanotransduction [13], and drug resistance [14]. Core components of the Hippo pathway encompass two serine/threonine (S/T) kinases (MST, the fly homolog of Hippo, and LATS), two adaptor proteins (MOB1 and SAV), and two transcriptional co-activators (YAP and its paralog TAZ). Upon activation by various stimuli such as cell–cell contact, mechanical force, or nutrient deprivation, MST activates and phosphorylates LATS kinase. Activated LATS, in turn, phosphorylates YAP/TAZ at serine 217 (S127)/S89 residues, preventing their nuclear translocation and transactivation of downstream genes, including *CTGF* and *Cyr61*, in collaboration with the transcription factor TEAD (Figure 1) [15,16,17,18,19].

The dysregulation of the Hippo pathway, marked by the activation of oncogenic *YAP/TAZ* and loss of the tumor suppressor gene *LATS*, has been implicated in various human cancers, including breast and lung cancers. Consequently, targeting the Hippo pathway has emerged as a crucial strategy for cancer therapy [20].

While the reversible addition (phosphorylation) and removal (dephosphorylation) of phosphate by kinases and phosphatases, respectively, to proteins or lipids is recognized as a pivotal post-translational modification in signal transduction across various cellular processes [21,22,23,24], existing research predominantly focuses on the roles of S/T kinases (e.g., PI3K, MAPK, TAOKs) and phosphatases (e.g., PP1, PP2A, STRPAK) in regulating the Hippo pathway in tumorigenesis [25]. The involvement of tyrosine phosphorylation and dephosphorylation in Hippo pathway regulation remains largely unexplored. Recent studies, including our own, have identified the Hippo pathway as a central mediator of receptor tyrosine kinase (RTK)-induced tumorigenesis, angiogenesis, immune evasion, and metastasis (Figure 1) [26,27,28,29,30]. Although several protein tyrosine kinases (PTPs) such as PTPN14 and PTPN21 have been reported as regulators of the Hippo pathway (Figure 1) [31,32,33,34], the systematic exploration of how to regulate the Hippo pathway has been lacking. In this study, we conducted a gain-of-functional screen and identified several novel PTP regulators of the Hippo pathway. Specifically, we validated PTPN12 as a novel negative regulator of the Hippo pathway effectors YAP/TAZ, influencing cell proliferation and migration.

## 2. Results

### 2.1. Gain-of-Functional Screening for PTPs Regulating the Hippo Pathway

In order to screen for novel phosphatases regulating the Hippo pathway, we cotransfected constructs expressing the *LATS* kinase biosensor (*LATS-BS*) or *YAP/TAZ* transcriptional co-activating reporter (*STBS-Luc*) alone or together with a PTP from a library of 68 PTPs that we previously established [34]. The luciferase activities of the LATS-BS or STBS-Luc reporter were measured. Significantly, 25 PTPs increase LATS-BS activity more than 2.5-fold, while 5 PTPs decrease LATS-BS activity more than 2.5-fold (Figure 2A). Given the opposing roles of LATS and YAP/TAZ in tumorigenesis [29], in contrast to the effect of PTPs on LATS-BS, most phosphatases decreased the signal of the reporter compared to the control. In total, 8 PTPs increase STBS-Luc more than 2-fold, whereas 19 PTPs decrease STBS-Luc activity more than 2-fold (Figure 2B). A total number of 38 PTP candidates were selected for further validation through a triplicate test. All 38 candidates were divided into three different groups. The first group was the PTPs that either increased or decreased the signal of LATS-BS. The second group included PTPs that had a significant effect on the signal of the STBS-Luc reporter. The third group were phosphatases that have significant opposite effects on the signals of *LATS* biosensors and the *YAP/TAZ* reporter. Then, a triplicate test was performed to validate the effects of PTPs on the signals of LATS-BS and STBS-Luc (Figure 3). Next, 25 PTPs were validated for PTPs regulating LATS-BS, whereas 16 PTPs were validated for PTPs regulating STBS-Luc. Of the 28 candidates regulating the signal of LATS-BS or STBS-Luc, several PTPs including *PTPN11*, *PTPN14*, *PTPN21* and *CDC14* were previously shown to be PTPs regulating the Hippo pathway [32,35,36,37,38,39], suggesting that the gain-of-functional assay worked. Out of 28 validated *PTPs*, 4 *PTPs* (*PTPN12*, *PTPDC1*, *PTPRR* and *DUSP1*) were selected based on the novelty, function, and fold change for further experiments.

### 2.2. Validation of Candidate PTPs Regulating the Hippo Pathway

All previous experiments were performed on phosphatases regulating the Hippo pathway by using LATS-BS and STBS-Luc. All of the four PTPs regulating the Hippo pathway were shown to increase LATS activity (LATS-BS) and decrease YAP/TAZ activity (STBS-Luc reporter). In other words, the phosphatases were associated with increased phosphorylation of YAP and TAZ at S127 and S89, respectively, which led to the sequestration of YAP/TAZ in the cytoplasm and interaction with the 14.3.3 protein. To validate the PTP candidates regulating the Hippo pathway, HEK293T cell lines stably expressing inducible PTPs were first stablished via lentiviral infection. Western blot analysis shows that PTP overexpression after doxycycline (Dox) induction causes an increased level of YAP phosphorylation on S127 (good TAZ-phospho-S89 is not available) compared to the cells without Dox treatment (Figure 4), providing further confirmation that the candidate phosphatase can regulate the Hippo pathway by increasing endogenous YAP phosphorylation and preventing it from translocating to the nucleus.

### 2.3. Validation of PTPN12 as a Novel Regulator of the Hippo Pathway

#### 2.3.1. Effects of PTPN12 Overexpression on the mRNA Expression Level of Hippo-Targeted Genes in HEK293T-PTPN12

Of a total number of four candidates, we selected PTPN12 for a further experiment and validation because it has previously been shown to play an important role in breast cancer initiation and progression [40,41], which is similar to the role of the Hippo pathway in breast cancer [17,19]. Since PTPN12 increases the signal of LATS-BS and stimulates the phosphorylation of endogenous YAP at S127 (Figure 4), it may result in the sequestration of YAP into the cytoplasm, which inhibits its transcriptional coactivating function, resulting in the downregulation of its target genes such as *CTGF* and *Cyr61*.

To examine the effects of PTPN12 on the expression level of the downstream genes of the Hippo pathway, the mRNA expression of these genes was assessed using qRT-PCR. Total RNA was extracted from the HEK293T-PTPN12 stable cell line with or without Dox treatment. qRT-PCR shows that the mRNA levels of both *Cyr61* and *CTGF* were downregulated around 2-fold by PTPN12, suggesting that *PTPN12* is a novel regulator of the Hippo signaling pathway (Figure 5).

#### 2.3.2. Effect of Loss-of-PTPN12 on Hippo Signaling and Cell Proliferation in Mammary Cells

The downregulation and loss-of-function mutations of *PTPN12* were shown in a variety of cancers, including breast cancer [40,42,43]. Thus, in this project, we aimed to identify the potential effects of *PTPN12* downregulation on tumorigenesis via the Hippo pathway. To further validate the role of *PTPN12* in regulating the Hippo pathway in breast cancer, *PTPN12* was knocked out in the human mammary epithelial cell line, MCF10A, by the CRISPR system. To improve gene knockout efficiency in a mixed cell population, a Dox-inducible system was used [44]. The Dox-inducible Cas9 endonuclease was first introduced into MCF10A cells via lentiviral infection and geneticin selection. As a green fluorescent protein (GFP) gene is located downstream of the *Cas9* promoter, so the expression of GFP induced by Dox was considered to be an indicator of the expression of our gene of interest. Therefore, GFP-high MCF10A-iCas9 cells were sorted after Dox treatment by flow-activating cell sorting (FACS). MCF10A-iCas9 was subsequently infected by lentivirus expressing guide RNAs to target two different regions of *PTPN12* (*gPTPN12-1, 2*) to establish Dox-inducible MCF10A-PTPN12-KO stable cell lines (Figure 6). Western blot analysis shows that the phosphorylation of YAP at S127 decreased in PTPN12-KO (+Dox) compared to the wild-type (−Dox) MCF10A cells (Figure 6A), further confirming that PTPN12 regulates the Hippo pathway by stimulating the S127 phosphorylation of the Hippo major effector protein YAP.

Previous studies show that upon phosphorylation at S127, YAP should be sequestered in the cytoplasm, which prevents it from translocating to the nucleus [18]. While YAP is localized in both the nucleus and cytoplasm in MCF10A-PTPN12-WT cells, it is more localized in the nucleus in the PTPN12-KO cells (Figure 6B). The quantitative analysis of YAP staining revealed that YAP was localized in the nucleus in more than 50% of MCF10A-PTPN12-KO cell lines, while only less than 25% of WT cell lines showed YAP nuclear localization. Overall, these findings indicate that YAP localization is highly affected by PTPN12, which validates the result of the previous experiment and confirms the potential role of PTPN12 in the regulation of the Hippo pathway.

The findings of the present study showed that PTPN12 regulates the Hippo pathway via the stimulation of YAP phosphorylation at S127, which prevents it from translocating to the nucleus to transactivate downstream gene transcription. YAP has oncogenic activity in the signaling pathway, and hyperactivation of the transcription coactivator can lead to an increase in cell proliferation. Since *PTPN12* KO enhances YAP nuclear localization, it may lead to increased cell proliferation mediated by enhanced activity of YAP. To examine the role of *PTPN12* in cell proliferation, MCF10A-PTPN12-KO (Figure 7A) and WT cells were cultured and counted for several days (Figure 7B). It is clear that PTPN12-KO causes a more significant increase in cell proliferation compared to the WT. To investigate if the increased cell proliferation in MCF10A-PTPN12-KO was due to activation of *YAP* and its paralog *TAZ*, *YAP* and *TAZ* were knocked down by siRNAs in MCF10A-gPTPN12 cells (Figure 7A). The knockdown of YAP and TAZ by siYAP and siTAZ in MCF10A-PTPN12-KO significantly reversed loss-of-PTPN12-induced increased cell proliferation (Figure 7B). These findings strongly suggest that the loss of *PTPN12* upregulates cell proliferation by activating the Hippo pathway effectors and oncoproteins YAP and TAZ.

#### 2.3.3. Effect of PTPN12 Overexpression on Hippo Signaling and Cell Proliferation and Cell Migration in Mammary Cells

Finally, given *PTPN12*′s role as a tumor suppressor gene, we aimed to investigate whether its overexpression could mitigate the tumorigenic and metastatic effects induced by the oncoproteins YAP/TAZ, as previously reported [45,46]. YAP- or TAZ-overexpressing MCF10A cells were transduced with lentivirus expressing Dox-inducible PTPN12 (Figure 8). Consistent with results obtained in HEK293T cells (Figure 4), the overexpression of PTPN12 (+Dox) led to an increase in YAP-pS127 (Figure 8). Considering that YAP/TAZ overexpression induces tumorigenic and metastatic phenotypes, such as enhanced cell proliferation and migration in MCF10A mammary cells [15,26,47], we evaluated whether PTPN12 could counteract these effects. Remarkably, PTPN12 overexpression resulted in a reduction in YAP/TAZ-induced increased cell proliferation (Figure 8A,B) and cell migration (Figure 9A–D) in MCF10A mammary cells.

In addition, the overexpression of PTPN12 (Figure 10A) inhibits TAZ-induced increased cell proliferation (Figure 10B) and anchorage-independent growth on soft agar (Figure 10C,D), as well as promoting cell migration (Figure 11A,B) in NMuMG mouse mammary cells.

## 3. Discussion

### 3.1. Identification of Novel PTP Regulating the Hippo Pathway

To identify the phosphatases regulating the Hippo pathway, a gain-of-functional screen with 68 PTPs was performed using LATS-BS and STBS-Luc reporter. In total, 28 PTP candidates were found to be involved in the regulation of the Hippo pathway as they had significant effects on the signals of STBS-Luc or LATS-BS or both of them. Previous studies demonstrate that *PTPN11*, *PTPN14*, *PTPN21*, and *CDC14* are involved in the Hippo pathway regulation [25]. Thus, in the current study, we identified 24 novel PTPs regulating the Hippo pathway through the two parallel screenings. The fold changes in the signal of the LATS biosensor by the PTP candidates were significantly higher than the fold change in the signal of STBS-Luc, which is mainly because of the high sensitivity of LATS-BS. These screenings suggested that the LATS biosensor and YAP/TAZ reporter used in this study are sensitive tools able to identify novel regulators of the Hippo pathway in real time. Based on the function and fold change in the signal of biosensor and reporter, *PTPRR*, *DUSP1*, *PTPDC1*, and *PTPN12* were selected for further validation.

PTPRR, a transmembrane tyrosine phosphatase, is inactivated in various malignancies, impacting tumorigenesis [48,49]. The loss of *PTPRR* in PTPRR-deficient mice leads to the hyperphosphorylation of ERK, affecting MAPK signaling [50,51]. *PTPRR* downregulation is reported in ovarian cancer; its re-expression reduces cell proliferation, inhibiting tumorigenesis [51,52]. Studies suggest that PTPRR negatively regulates cell proliferation in cancer cells, prompting an intriguing evaluation of its potential role in the Hippo pathway in tumorigenesis.

DUSP1, a dual-specific phosphatase, regulates MAPK by dephosphorylating ERK, contributing to tumorigenesis [52,53,54,55]. In hepatocellular cancer cells, DUSP1 expression inversely correlates with phosphorylated ERK and cell proliferation [56,57]. The downregulation of DUSP1 alters gene expression in various pathways, including metastasis, MAP kinase activity, and RTK activity [58]. Despite its oncogenic role in several cancers [59,60,61,62], the overall function of DUSP1 in tumor progression remains inconclusive. Given its dual role in tumorigenesis, exploring the crosstalk between the Hippo pathway and DUSP1 is intriguing. Considering the tumor suppressor roles of DUSP1 mediated by MAPK, and MAPK’s involvement in the Hippo pathway regulation, DUSP1 may regulate the Hippo pathway either via direct dephosphorylation of the Hippo component or indirectly through MAPK regulation.

*PTPDC1* (protein tyrosine phosphatase domain containing 1) is one of the four members of *CDC14s* phosphatase family [63]. Until now, there has not been any study of the role of PTPDC1 in cancer. As PTPDC1 comprises the critical domains of DSPc (dual-specificity phosphatase, catalytic domain), it is thought that PTPDC1 may be involved in tumorigenesis [64].

Therefore, it will be interesting to further evaluate how PTPC1, PTPRR, and DUSP1 regulate the Hippo pathway in tumorigenesis and metastasis.

### 3.2. PTPN12 Tumor Suppressor Function in Cancer Is Mediated by the Hippo Pathway

Due to its pivotal role in breast cancer and its function as a master regulator of RTKs [40,41,65,66], we investigated how PTPN12 influences tumorigenesis via the Hippo pathway. PTPN12 is implicated in various biological functions such as cell migration, adhesion, immunity, and survival [41,67,68]. The dysregulation of the Hippo pathway, through mutation or altered expression of its components, disrupts cell contact inhibition, leading to uncontrolled cell proliferation—a hallmark of oncogenic transformation [16]. Therefore, assessing the crosstalk between PTPN12 and the Hippo pathway is crucial for understanding the underlying mechanisms of PTPN12 loss in tumorigenesis. In line with this, we demonstrated that the loss of PTPN12 in MCF10A cells increased cell proliferation in a Hippo-dependent manner. This excessive cell proliferation, a critical process in tumorigenesis due to PTPN12 loss, is mediated by the Hippo signaling pathway effectors YAP and TAZ. Furthermore, we provided compelling evidence that PTPN12 exerts its tumor suppressor function through the Hippo pathway by suppressing YAP/TAZ-induced increased cell proliferation and migration. Our findings strongly suggest that the PTPN12-YAP/TAZ signaling axis may play a crucial role in mammary tumorigenesis and metastasis.

While serine phosphorylation, such as the phosphorylation of YAP S127, is crucial for regulating the core components of the Hippo pathway, recent studies have highlighted the involvement of tyrosine phosphorylation in Hippo pathway regulation. We have recently show that RTKs, including EGFR, VEGFR, FGFR and RET, play significant roles in modulating the Hippo signaling pathway in cell proliferation, cell migration, and angiogenesis [26,29]. In breast cancer, recent findings also indicate that the loss of PTPN12 phosphatase function activates several oncogenic RTKs such as MET, PDGFRb, HER2, and EGFR [40,41,66], positioning PTPN12 as a master regulator of protein tyrosine kinases. PTPN12 phosphatase function is compromised in breast cancer by inactivating mutations, deletions, or loss of expression, leading to the overactivation of multiple RTKs, particularly in triple-negative breast cancer (TNBC) subsets. Therefore, either restoring PTPN12 expression or employing a combinatorial inhibition of PTPN12-regulated RTKs in PTPN12-deficient TNBC cells inhibits cell proliferation [40,66]. As the oncogenic function of RTKs was associated with the increased oncogenic activity of YAP and TAZ through either the direct or indirect regulation of YAP/TAZ activity by the tyrosine kinases [26,29], further studies need to investigate how PTPN12 is connected to the Hippo pathway. Tyrosine phosphatases such as PTPN12 may either directly regulate YAP and TAZ via the dephosphorylation of YAP and TAZ at tyrosine residues or via the indirect regulation of YAP and TAZ mediated by RTKs and RTK/MAPK/PI3K. Understanding the underlying molecular mechanism of the Hippo pathway regulation by PTPN12 may provide information about targeting oncogenes that are overactivated upon the loss of PTPN12 in cancer treatment. Furthermore, considering the contrasting roles of PTPN12 and YAP/TAZ in breast cancer, it would be intriguing to investigate whether there is a reverse correlation between the levels of PTPN12 and YAP/TAZ, as well as if this correlation is linked to poor survival outcomes in breast cancer patients.

## 4. Materials and Methods

### 4.1. Plasmid Construction and Purification

pTRIPZ lentiviral vector was used to construct all of the Dox-inducible PTP expression plasmids. For plasmid construction, cDNAs of PTPN12, PTPRR, PTPDC1, and DUSP1 previously constructed by us [30] were first amplified via polymerase chain reaction (PCR) using the gene-specific primers shown in Table 1. PCR was performed using PrimeSTAR GXL DNA polymerase (Takara, Mississauga, ON, Canada) according to the manufacturers’ instruction. All PCR products and vector were digested by AgeI and MluI restriction enzymes, followed by the subcloning of the digested PCR product into the AgeI/MluI sites of pTRIPZ vector. Plasmids were purified using the QIAprep Spin Miniprep Kit (Qiagen, Montreal, QC, Canada) according to the manufacturer’s protocol.

To knock out the *PTPN12* gene in MCF10A, 2 different guide RNA (gRNAs) sequences targeting *PTPN12* (gPTPN12) were chosen from the data set generated by David Root’s group in 2016 [69]. The gRNAs were optimized with maximum on-target effects and minimum off-target activity: gPTPN12-1: forward, 5′-CACCGTTTGTGCCATAGATTATACG-3′; reverse, 5′-AAACCGTATAATCTATGG-CACAAAC-3′; gPTPN12-2: forward, 5′-CACCGAAGAAGGTCCCTCTCCAAGA-3′; reverse, 5′-AAACTCTTGGAGAGGGACCTTC TTC-3′. A BsmBI overhang sequence was added to each of the single-stranded complementary oligoes when they were synthesized. The Lentiviral LentiGuide-Puro vector was digested by the Esp3I/BsmBI fast digest enzyme (Thermo Fisher, Nepean, ON, Canada, # FD0454). The digested vector was separated on agarose gel and purified via the QIAquick Gel Extraction Kit (Qiagen) and eluted in EB buffer. To anneal and phosphorylate oligos, T4 polynucleotide kinase (PNK) (NEB, Whitby, ON, Canada # M0201S) and 10× T4 ligation buffer (NEB, # B0202S) were added to each tube containing the gPTPN12-F and gPTPN12-R primers. Following the addition of PNK and buffer, the oligo tubes were put in a thermocycler running at 37 °C for 30 min, 95 °C for 5 min, and then ramped down to 25 °C at 5 °C per min. Then, the ligation reaction was catalyzed by mixing the annealed oligos andEsp3I/BsmBI-digested LentiGuide-Puro vector with Blunt/TA ligase master mix (NEB, # M0367S), followed by a heat shock transformation into a Stbl3 competent cell.

### 4.2. Cell Culture

HEK293T (human embryonic kidney; ATTC, Burlington, ON, Canada, Cat# CRL-3216) and HEK293AD (Cell Biolabs, Burlington, ON, Canada, LLC, Cat#AD-100) cell lines were maintained in Dulbecco’s Modified Eagle’s Medium (DMEM; Sigma-Aldrich, Oakville, ON, Canada, # D6429) supplemented with 10% fetal bovine essence (FBE) and 1% penicillin/streptomycin (P/S) (Invitrogen, Burlington, ON, Canada). MCF10A (human immortalized mammary epithelial) cells expressing inducible Cas9 (iCas9) were cultured in DMEM/Nutrient Mixture F12 Ham without L-glutamine (Sigma-Aldrich, Oakville, ON, Canada, # D6421) supplemented with 5% horse serum (HS) (Sigma-Aldrich, # H1270), 2.5 mM of L-Glutamine (Sigma-Aldrich, # G7513), 10 ng/mL of insulin (Sigma-Aldrich, # I6634), 20 ng/mL of human epidermal growth factor (hEGF) (Sigma-Aldrich, # E4269), 100 ng/mL of cholera toxin (Sigma-Aldrich, # C8052), 0.5 μg/mL of hydrocortisone (Sigma-Aldrich, # H4001), and 1% P/S. All cells were maintained in a 37 °C incubator with 5% CO_2_.

### 4.3. Lentivirus Production

Plasmid transfection for lentivirus production was performed using a Poly-jet transfection reagent according to the manufacturer’s protocol (SignaGen Laboratories, Toronto, ON, Canada). In brief, 1 × $10^{6}$ HEK293T cells (passage number lower than 10) were seeded in each 35 mm cell culture plate pre-coated with 250 µL of 0.1 mg/mL poly-L-lysine. The next day, the growth media in each plate were replaced with prewarmed fresh media 30 min prior to transfection. Then, 0.5 µg of each of the pTRIPZ-PTPs, or Lenti-iCas9-neo or gPTPN12 in LentiGuide plasmid, was mixed with 0.38 µg of psPAX viral packaging plasmid, 0.13 µg of pMD2G viral envelop plasmid, and 3 µL of PolyJet transfection reagent (SignaGen, # SL100688) in serum-free DMEM medium and incubated at room temperature for 15 min. The mixture was added dropwise into the medium containing cells. The next day, the medium was replaced with 1 mL of DMEM/10%FBS containing 10 mM of Sodium Butyrate (Santa Cruz Biotechnology, Santa Cruz, CA, USA # sc-202341A). Sodium Butyrate induces gene expression in cells. About 24 h after the treatment of the cells with Sodium Butyrate, the lentivirus-containing media were collected and aliquoted into 250 µL per 1.5 mL Eppendorf tube. After flash-freezing in liquid nitrogen, all virus aliquots were kept at −80 °C.

### 4.4. Establishment of Stable Cell Lines

To establish HEK293T cells overexpressing PTPs, 3 × 10^5^ HEK293T cells were seeded in each well of a 12-well plate. One day after seeding (~40–50% confluency), cells were infected with varying amounts of lentivirus (0, 10, 25, 50, 100, 250 µL) expressing doxycycline (Dox)-inducible PTPs or Cas9 in the presence of 8 µg/mL of polybrene in a 500 µL/well. Twenty-four hours post-infection, cells were subjected to selection with 1.2 of µg/mL puromycin.

To establish MCF10A cells expressing Dox-inducible gPTPN12, MCF10A cells were first infected with lentivirus expressing iCas9, as described above, following by selection at 400 µg/mL G481 (neomycin). The established MCF10A-iCas9 cells were further selected by selecting top 15% GFP-positive cells by FACS 2 days after the Dox induction of GFP and Cas9.

The sorted cells were subsequently infected with lentivirus expressing gPTPN12, followed by puromycin selection at 2.5 µg/mL. Finally, the stable cell line was treated with 1ug/mL Dox for five days to induce Cas9 expression that leads to PTPN12 KO.

To establish inducible PTPN12 in MCF10A or NMuMG cells overexpressing wild-type YAP or TAZ, previously established YAP/TAZ overexpressing cells [15,70] were infected with lentivirus expressing PTPN12 cloned in a Dox-inducible vector pTRIPZ, followed by puromycin selection at 1.5 µg/mL.

### 4.5. Protein Extraction and Western Blot Analysis and Antibodies

Protein extraction and Western blot analysis were carried out as described previously [15]. The band intensities of YAP and phosphorylated YAP (pYAP) were quantified using ImageJ. The antibodies in Table 2 were used in our studies (Table 2).

### 4.6. RNA Extraction and qRT-PCR Analysis

RNA extraction and qRT-PCR were carried out as described previously [71]. The primer sequences for qRT-PCR analysis were as follows: (1) *Cyr61*: forward, 5′-AATGGAGCCTCGCATCCTATA-3′; reverse, 5′-TTCTTTCACAAGGCGGCA-3′; (2) *CTGF*: forward, 5′-CCCTCGCGG CTTACCGACTGG-3′; reverse, 5′-CACAGGTCTTGGAACAGGCGC-3′.

### 4.7. Luciferase Assays and Gain-of-Functional Screening

To evaluate the activities of LATS kinase and YAP/TAZ, LATS-BS and STBS-Luc reporter were used, respectively, in luciferase assays. To validate LATS-BS and STBS-Luc, 3 × $10^{5}$ HEK293T cells were plated in each well of a 12-well plate. One day after cell seeding, 100 ng of LATS-BS and STBS-Luc plasmids were transfected alone or together with LATS2/MST2 or TAZ, respectively, into HEK293AD cells using the Polyjet transfection reagent (SignaGen, # SL100688) described in Section 2.3. One day after transfection, cells were lysed in 150 µL of passive lysis buffer (PLB, Promega, Madison, WI, USA, # E194). Luciferase activity of STBS-Luc reporter was measured via the Luciferase Assay Kit (Promega, # E1910) using a Turner Biosystems 20/20 luminometer, whereas that of LATS-BS was measured via the Nano-Glo^®^ Luciferase Assay System (Promega, #N1110). The fold changes were calculated as the ratio of relative light unit (RLU) of LATS-BS + LATS2/MST2 or STBS-Luc + TAZ to that of LATS-BS or STBS-Luc, respectively. The data are the mean ± standard deviation (S.D.) of the triplicate samples. The experiments were repeated twice.

For the gain-of-functional screen, a library including 68 PTPs previously constructed by us was used [34]. LATS-BS or STBS-Luc plasmids were either transfected into HEK293AD alone or together with each PTP from the PTP library. To perform the cotransfection of PTPs and LATS-BS, a total amount of 500 ng plasmids, including 100 ng of SmBiT-14-3-3, 100 ng of Lg-BiT-YAP, and 300 ng of PTP, was transfected into the cells in each well of a 12-well plate. With regard to the cotransfection of PTP and STBS-Luc, 100 ng of STBS-Luc plasmid was cotransfected with 400 ng PTP. One day after transfection, cells were lysed in 150 µL of PLB, followed by the luciferase assays described above. Fold change was calculated via the ratio of the RLU of LATS-BS or STBS-Luc plus PTP to that of the control (RLU of LATS-BS or STBS-luc alone). The candidate PTPs were further validated via the transfection of triplicate samples, followed by luciferase assays, as described above.

### 4.8. Immunostaining

MCF10A-PTPN12-KO and MCF10A-WT Cells were diluted into 4 × $10^{4}$ cells per mL and dispensed 0.5 mL into each well of a 24-well plate. Before seeding the cells, one coverslip was placed in each well of a 24-well plate. After two days of cell seeding, cell staining was performed to evaluate and compare the localization of YAP between the MCF10A-PTPN12-KO and MCF10A-WT cell lines. For immunostaining, cells were fixed in 200 µL of a 4% Paraformaldehyde (PFA) at RT for 10 min, followed by cell permeabilization with 200 µL of 0.2% Triton X-100 in 1× PBS at RT for 2 min. After permeabilization, blocking was performed by adding 200 µL of blocking and hybridization solution (BHS; 10% bovine serum albumin and 5% normal goat serum) and incubated on a shaker for 40 min. Cells were incubated in the Alex Fluor 488 conjugated anti-YAP antibody (D8H1X) XP^®^ Rabbit mAb (NEB, #14074, 1:100) on a shaker with gentle agitation at RT for 1 h in the dark. DAPI was used to stain DNA in cells. An Inverted Nikon Eclipse TE 2000U microscope (Nikon, Mississauga, ON, Canada) was used to capture fluorescence images.

### 4.9. Transient Knockdown of YAP and TAZ in MCF10A-PTPN12-KO

Small interfering RNAs (siRNAs) duplex targeting *YAP* or *TAZ* (siYAP or siTAZ) were purchased from Integrated DNA Technologies (IDT, Coralville, IA, USA). A day before transfection with siRNAs, 4 × 10^5^ cells were seeded in a 35 mm plate. MCF10A-PTPN12-KO cells were transfected with 100 nM of siYAP and siTAZ using a GenMute siRNA transfection reagent (SignaGen, SL100568) based on the manufacturer’s protocol. Five hours after transfection, cells were collected for further experiments such as cell proliferation assay. The efficiency of the siRNAs in the knocking down of YAP and TAZ was determined via Western blot analysis three days after transfection.

### 4.10. Cell Proliferation Assay

About 2 × 10^4^ cells were plated in triplicate into each well of a 24-well plate. Cell counting was performed on days 2, 4, and 6 after cell seeding. The media in each well were refreshed every 3 days via replacement with complete growth media.

### 4.11. Soft Agar Assay

Triplicates of 2 × 10^4^ cells were suspended in a 0.4% low melting point agarose in DMEM and overlayed on a 0.8% base agarose in DMEM. Plates were incubated in a 37 °C incubator. Fresh DMEM was added into each plate every 3 days. Colony formation could be observed 3 weeks later. Colonies were stained in 0.005% of crystal violet solution for 1 h and quantified using ImageJ. The average and standard deviation of triplicate samples were calculated.

### 4.12. Cell Migration Assay

Wound healing analysis was used for cell migration assay. In brief, 2 × 10^4^ cells were seeded into each well of a 96-well plate. Cells were grown until confluent next day. Wounds were made using a WoundMaker^TM^ (Perkin Elmer, Vaughan, ON, Canada). The closure of wounds was monitored for 72 h using a Incycytes Zoom Live Cell Analysis system (Sartorius, Oakville, ON, Canada) at 37 °C in an incubator. The percentage of wound closure was calculated for each time point. The experiments were repeated twice. The means and standard deviations (S.D.) of triplicate samples for each time point were calculated.

## 5. Conclusions

In summary, this study has uncovered numerous novel PTPs influencing the Hippo pathway through gain-of-functional screening. It presents compelling evidence that PTPN12 modulates cell proliferation by affecting the Hippo effector proteins YAP and TAZ, suggesting a potential strategy for treating breast cancer patients with compromised PTPN12. Given PTPN12’s involvement in other cancer types, further exploration of its interactions with the Hippo pathway in diverse cancers is warranted. Validating these findings across various cell lines and cancer types would enhance our understanding of PTPN12’s tumor suppressor role. Deciphering the functional significance of PTPs and the Hippo signaling pathway could yield novel insights into cancer initiation, progression, and development.

The preliminary data from this study can guide future projects, particularly in validating over 10 identified PTP candidates that regulate the Hippo pathway. Investigating the role of these PTPs in Hippo pathway regulation and exploring their impact on tumorigenesis or metastasis represents an intriguing avenue for further research.

## Figures and Tables

**Figure 1 ijms-25-04064-f001:**
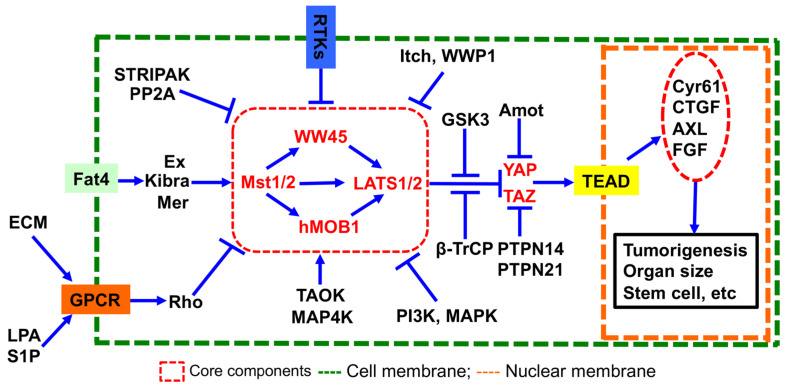
The mammalian Hippo pathway and its regulators. LPA, lysophosphatidic acid; GPCR, G protein-coupled receptor; ECM, extracellular matrix.

**Figure 2 ijms-25-04064-f002:**
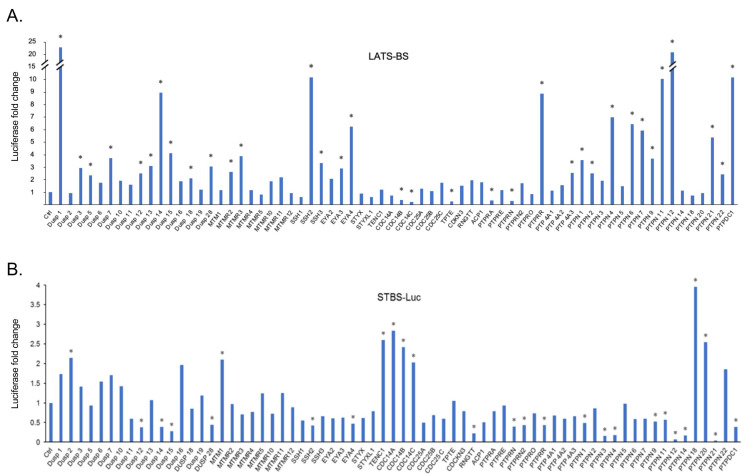
Parallel screens for PTPs regulating the Hippo pathway by the LATS-BS and STBS-Luc (YAP/TAZ reporter). The ratio of the LATS-BS (**A**) and STBS-Luc reporter (**B**) signals after cotransfection with phosphatases to that of LATS-BS or STBS-Luc alone was calculated based on fold changes. A fold change greater than 2 was considered significant and indicated as “*”.

**Figure 3 ijms-25-04064-f003:**
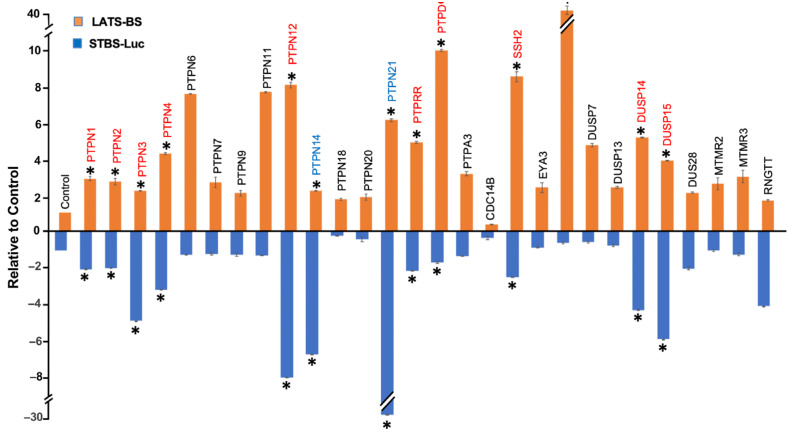
Validation of candidate PTPs regulating the Hippo pathway. A triplicate test was performed to validate the result of the first screen. The fold changes in LATS-BS and STBS-Luc reporters were calculated after cotransfection with each phosphatase. “*” represents a fold change greater than 2.

**Figure 4 ijms-25-04064-f004:**
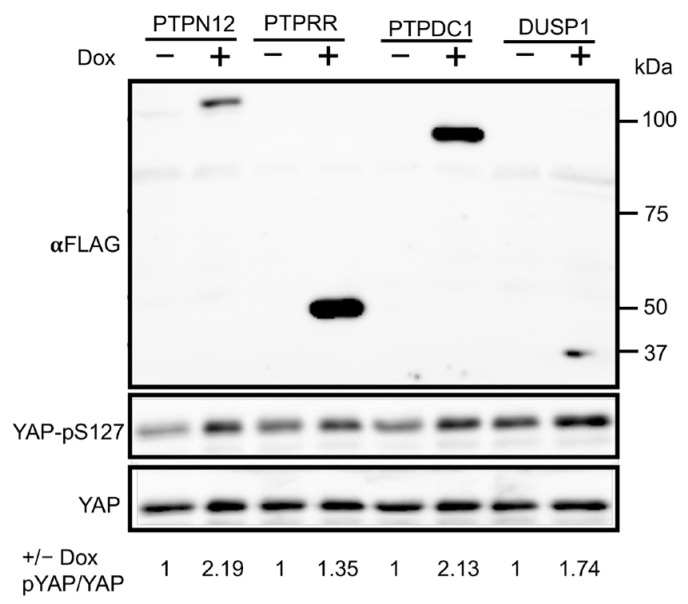
PTP candidates stimulate phosphorylation of YAP-pS127 endogenously. Western blot analysis of YAP-pS127 after induction of PTP candidate expression by Dox in HEK293T stable cell line. Without Dox, HEK293T-PTP stable cell lines do not express PTPs, while Dox treatment induces expression of FLAG-tagged PTPs. Band intensity of YAP and phosphorylated YAP-pS127 was quantified via ImageJ 1.54g. Expression of each of PTP candidates stimulates phosphorylation of YAP at S127, which is indicated by intensity ratio of ±Dox.

**Figure 5 ijms-25-04064-f005:**
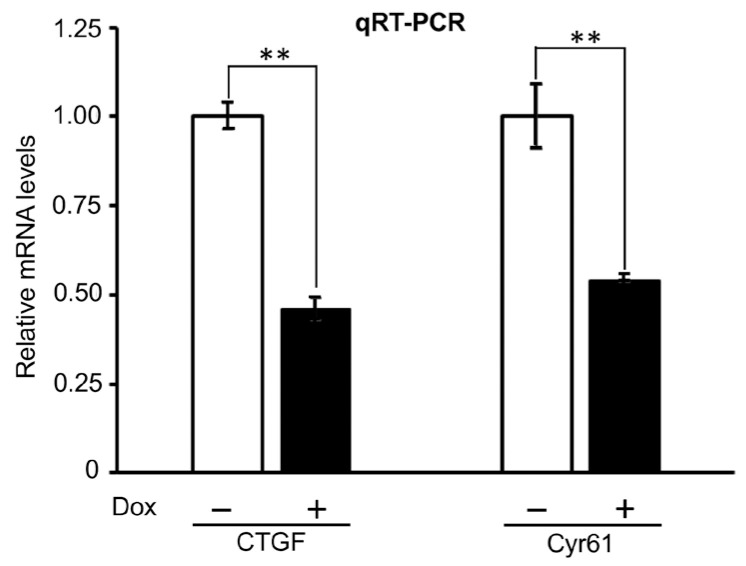
PTPN12 overexpression downregulates the mRNA expression of CTGF and Cyr61. qRT-PCR analysis of mRNA levels of CTGF and Cyr61 in HEK293T-PTPN12 stable cell line in the absence (−) and presence (+) of Dox treatment. The relative mRNA levels were calculated based on the ratio of mRNA levels under ± conditions. The mean ± S.D. of the triplicate samples are shown. **, *p* ≤ 0.01 (statistical significance).

**Figure 6 ijms-25-04064-f006:**
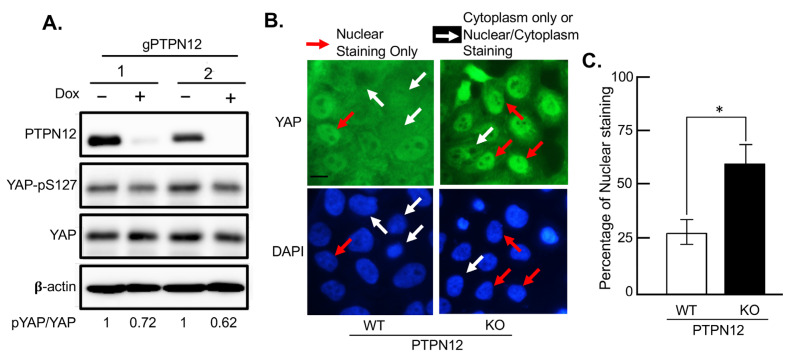
Loss of PTPN12 reduces phosphorylation of YAP at S127 and increases YAP nuclear localization. (**A**) Western blot analysis. Reduced phosphorylation of YAP at S127 by PTPN12 in MCF10A-iCas9-gPTPN12-KO mammary cells. MCF10A-iCas9 cells stably expressing gPTPN12-2 (1) or gPTPN12-3 (2) were left untreated (−) or treated (+) with Dox for 5 days, followed by Western blot analysis of PTPN12, YAP, and S127-phosphorylated YAP (YAP-pS127) using anti-PTPN12, anti-YAP, and anti-pS127-YAP antibodies. Band intensity of YAP and pYAP was quantified using ImageJ. Relative intensity of phosphorylated YAP (pYAP) and YAP (pYAP/YAP ratio) is shown. (**B**,**C**) Immunostaining analysis of YAP subcellular localization. YAP immunostaining in MCF10A cells with wild-type (WT) or knockout (KO) PTPN12. DAPI was used to stain nucleus. For each cell line, approximately 100 cells were counted for YAP nuclear-only (red arrow in (**B**)) or cytoplasmic/cytoplasmic + nuclear (white arrow in (**B**)) staining cells. Scale bar, 200 μm. Quantitative analysis of YAP staining showed that in more than 50% of MCF10A-PTPN12-KO cell lines, YAP was localized in nucleus, whereas less than 25% of WT cell lines demonstrated YAP nuclear localization (**C**). *, *p* ≤ 0.05 (statistical significance).

**Figure 7 ijms-25-04064-f007:**
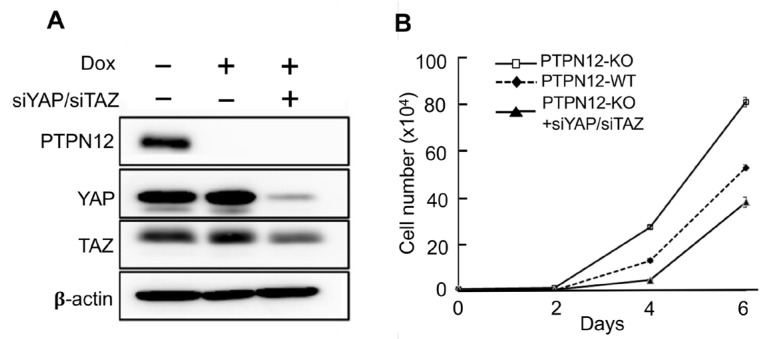
PTPN12 regulates cell proliferation, which is mediated by the Hippo signaling pathway. (**A**) Western blot analysis of YAP and TAZ after the siYAP/siTAZ transfection of short interference RNAs targeting YAP/TAZ (siYAP/siTAZ) into MCF10A-gPTPN12 cells. (**B**) Cell proliferation assay. Cell proliferation was measured in MCF10A-PTPN12-WT (−Dox, −siYAP/siTAZ), MCF10A-PTPN12-KO (+Dox, −siYAP/siTAZ), and MCF10A-PTPN12-KO-siYAP/siTAZ (+Dox, +siYAP/siTAZ) cells. Cell numbers were counted on days 2, 4, and 6 after seeding.

**Figure 8 ijms-25-04064-f008:**
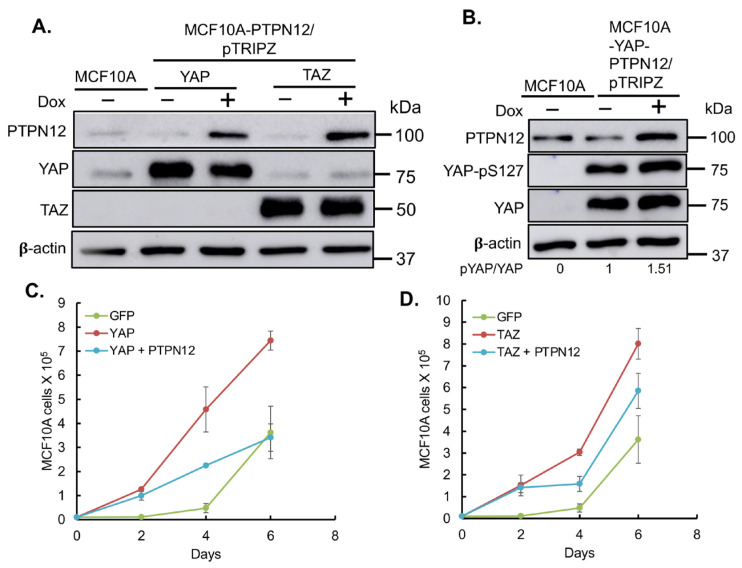
The overexpression of PTPN12 suppresses YAP/TAZ-induced increased mammary cell proliferation. (**A**) Inducible expression of PTPN in MCF10A-YAP or -TAZ cells. Dox was added to induce PTPN12 in MCF10A-YAP-PTPN12/pTRIPZ or MCF10A-TAZ-PTPN12/pTRIPZ cells. MCF10A cells expressing a WPI lentiviral vector expressing GFP were used as a control. Forty-eight hours after Dox induction, proteins were extracted from cells and subjected to Western blot analysis of PTPN12, YAP, and TAZ. β-actin was used as an internal control. (**B**) Increased levels of phosphorylated YAP after PTPN12 overexpression were noted. Protein lysates from A were subjected to Western blot analysis of PTPN12, YAP-pS217, and YAP. (**C**) The suppression of YAP-induced increased cell proliferation by PTPN12. MCF10A-GFP (WPI vector) and MCF10A-YAP-PTPN12 in the absence or presence of Dox were subjected to cell proliferation analysis. (**D**) The suppression of TAZ-induced increased cell proliferation by PTPN12. MCF10A-GFP (WPI vector) and MCF10A-TAZ-PTPN12 in the absence or presence of Dox were subjected to cell proliferation analysis. The mean and standard deviation (S.D.) of the cell numbers in triplicate samples at each day were shown.

**Figure 9 ijms-25-04064-f009:**
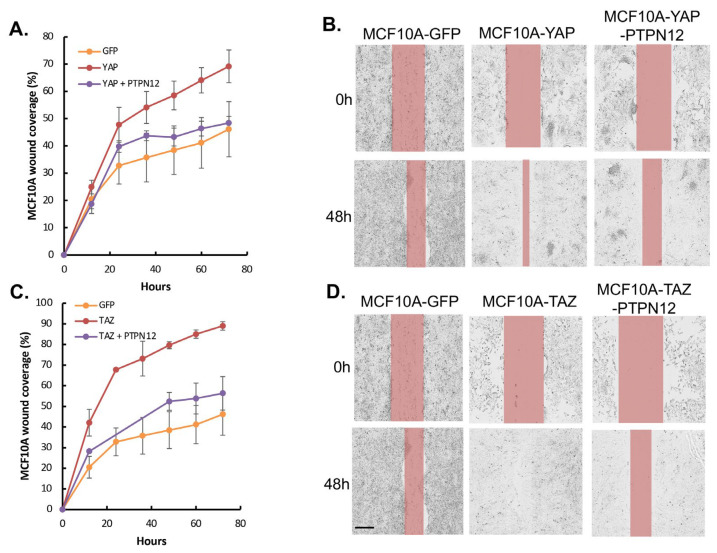
Inhibition of YAP/TAZ-induced increased cell migration by PTPN12 in MCF10A mammary cells. (**A**,**C**) Wound healing analysis using Incucyte Zoom. MCF10A-GFP (WPI vector) and MCF10A-YAP/TAZ expressing Dox-inducible PTPN12 cells in the absence or presence of Dox (+PTPN12) were seeded into 96-wells. After making the wounds, cell migration was monitored as a percentage of wound closure using Incucyte Zoom for 72 h. (**B**,**D**) Images from (**A**) showing wound closure at 0 h and 48 h. MCF10A-GFP was used as a control for both MCF10A-YAP/PTPN12 and MCF10A-TAZ/PTPN12. Scale bar, 300 μm.

**Figure 10 ijms-25-04064-f010:**
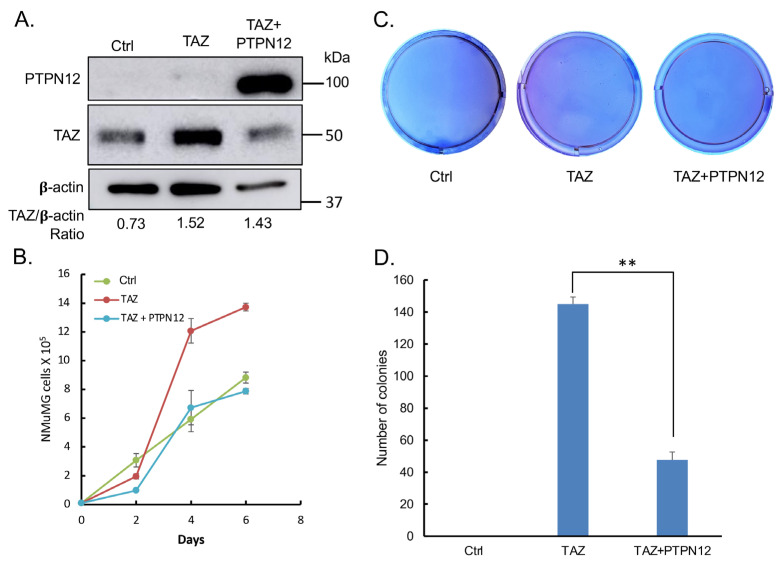
Inhibition of TAZ-induced increased cell proliferation and anchorage-independent growth on soft agar in NMuMG cells. (**A**) Western blot analysis. Protein was extracted from NMuMG [Control (Ctrl)] cells and NMuMG cells expressing TAZ in the absence (−Dox) or presence (+Dox) of PTPN12 in pTRIPZ. About 10 μg of protein lysates was subjected to Western blot analysis. Band intensity was quantified using ImageJ. The ratios of the intensity of TAZ and β-actin was calculated. (**B**) Cell proliferation analysis. (**C**,**D**) Soft agar colony formation analysis. Triplicates of 2 × 10^4^ cells for each cell line were mixed with top agarose and overlaid onto base agarose in each well of 6-well plates, followed by incubation at 37 °C for 3 weeks. Colonies were stained with crystal violet (**C**) and quantified (**D**) using ImageJ. **, *p* ≤ 0.01 (statistical significance).

**Figure 11 ijms-25-04064-f011:**
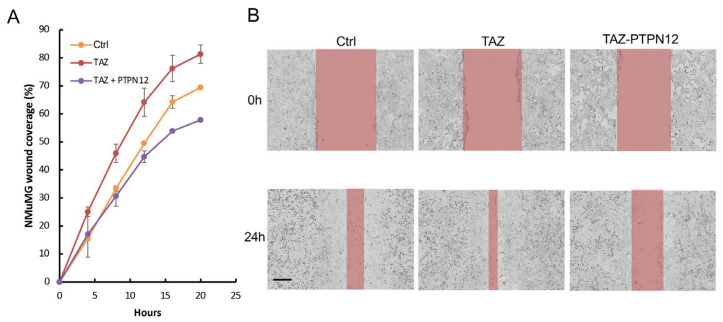
Inhibition of TAZ-induced increased cell migration by PTPN12 in NMuMG mammary cells. (**A**) Wound healing analysis using Incucyte Zoom. NMuMG (WPI vector) and NMuMG-TAZ expressing Dox-inducible PTPN12 cells in the absence or presence of Dox (TAZ-PTPN12) were seeded into 96-wells. After making the wounds, cell migration was monitored as a percentage of wound closure using Incucyte Zoom for 24 h. (**B**) Images from (**A**) showing wound closure at 0 h and 24 h. Scale bar, 300 μm.

**Table 1 ijms-25-04064-t001:** Gene-specific primers for plasmid construction.

Construct	Forward Primer (5′-3′)	Reverse Primer (5′-3′)
PTPN12-pTRIPZ	5′-CGACCGGTGCCACCATGGAAG ACTACAAAGACGATGACGACAA GATGGAGCAAGTGGAGATCCTG-3′	5′-ACGACGCGTTCATGTCCATTC TGAAGGTG-3′
PTPRR-pTRIPZ	5′-CGACCGGTGCCACCATGGACTA CAAAGACGATGACGACAAGATGA TTCTTCACAGATTAAAAGAAAG-3′	5′-ACGACGCGTTCACTGGACAGT CTCTGCTG-3′
DUSP1-pTRIPZ	5′-CGACCGGTGCCACCATGGACTA CAAAGACGATGACGACAAGATGG TCATGGAAGTGGGCAC-3′	5′-ACGACGCGTTCAGCAGCTGGG AGAGGTCGTAATG-3′
PTPDC1-pTRIPZ	5′-CGACCGGTGCCACCATGGACTA CAAAGACGATGACGACAAGATGG CTGCAGGAGTCTTGCC-3′	5′-ACGACGCGTCTAGAGGCCAGG CTTAGGGC-3′

**Table 2 ijms-25-04064-t002:** Antibodies used for Western blot analysis.

Primary or Secondary	Protein	Antibody	Dilution	Company
Primary	PTPs	α-FLAG (M2)	1:1000	Sigma-Aldrich
Primary	ꞵ-actin	α-ꞵ-actin	1:10,000	Sigma-Aldrich
Primary	Cas9	Cas9 (7A9-3A3) Mouse mAb	1:1000	Cell Signaling (Danvers, MA, USA)
Primary	PTPN12	PTP-PEST Rabbit mAb	1:1000	Cell Signaling
Primary	Phospho-YAP	Phospho-YAP (Ser127)-mAb	1:1000	Cell Signaling
Primary	YAP/TAZ	YAP/TAZ Rabbit mAb	1:1000	Cell Signaling
Secondary	-	HRP goat α-mouse	1:10,000	Cell Signaling
Secondary	-	HRP goat α-rabbit	1:10,000	Cell Signaling

## Data Availability

Data is contained within the article.

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
