# Peer review of "Identification of PTPN12 Phosphatase as a Novel Negative Regulator of Hippo Pathway Effectors YAP/TAZ in Breast Cancer"

_ijms, 2024, doi:10.3390/ijms25074064_

Round 1

Reviewer 1 Report

Comments and Suggestions for Authors

The authors did a thorough investigation of the protein tyrosine phosphatases (PTPs)’ effect on the regulation of the Hippo signaling pathway. In this manuscript, the authors screened 68 PTPs for their effect on LATS kinase biosensor (LATS-BS) and YAP/TAZ transcriptional co-activating reporter and validated that PTPN12 as a novel regulator of the hippo pathway and corresponding mechanism-of-action. Overall, the manuscript did a comprehensive investigation regarding the roles of PTPs in regulating the Hippo signaling pathway, especially in breast cancer cell line and will attract wide interest in this field. I suggest minor revisions. Here are some comments to the authors.

1.      I would suggest adding a graphical illustration at the beginning of the manuscript to describe the Hippo signaling pathway and corresponding suppressor/activator.

2.      Could the authors include a statistical analysis in Figure 4?

3.      Figure 5B is not clear enough to support the conclusion that nuclear localization of YAP increased in the PTPN12-KO cell line by itself. The scale bar is also missing.

4.      Please include more details in analyzing Figure 5B to achieve Figure 5C. How many cells were used in this analysis and how was the analysis performed?

5.      Is there any in vivo study or corresponding clinical study in this field? It would be great to include those in the discussion section. 

Reviewer 2 Report

Comments and Suggestions for Authors

In the present manuscript, Emami et al. demonstrate how PTPN12 negatively regulates Hippo pathway to influence breast cancer proliferation and migration. The study is novel; however, several modifications are required to improve the present version of the manuscript. My comments are given below: 

1. Why the role of PTPN14 is not checked?

2. The entire study is based on breast cancer, hence what is the point of use HEK293T cells (kidney cells) in certain areas?

3. Quantification of the blots in all the occasions is required for better understanding.

4. Figure 4: control missing. Cells without PTPN12 overexpression should be used.

5. Figure 5B: Not understood at all from the image. A better experiment would be to separate cytosolic fraction from the nucleus and analyze p-YAP and total YAP by western blotting.

6. Cell proliferation should be carried out by specific assays like BrdU incorporation assay or colony formation assay.

Comments on the Quality of English Language

Minor editing of English is required.

Round 2

Reviewer 2 Report

Comments and Suggestions for Authors

The authors adequately addressed all my concerns.

Author Response

Thank you very much